# AdipoRon Alleviates Liver Injury by Protecting Hepatocytes from Mitochondrial Damage Caused by Ionizing Radiation

**DOI:** 10.3390/ijms252011277

**Published:** 2024-10-20

**Authors:** Yi Liu, Yinfen Xu, Huilin Ji, Fenfen Gao, Ruoting Ge, Dan Zhou, Hengyi Fu, Xiaodong Liu, Shumei Ma

**Affiliations:** 1School of Public Health, Wenzhou Medical University, Wenzhou 325035, China; 15058752218@163.com (Y.L.); xyf2587@163.com (Y.X.); 15058719736@163.com (H.J.); 18895321604@163.com (F.G.); geruoting@163.com (R.G.); 17857333573@163.com (D.Z.); 19817563056@163.com (H.F.); 2South Zhejiang Institute of Radiation Medicine and Nuclear Technology, Wenzhou 325035, China; 3Zhejiang Provincial Key Laboratory of Watershed Science and Health, Wenzhou Medical University, Wenzhou 325035, China

**Keywords:** AdipoRon, mitochondrial damage, ionizing radiation, radiation protection, liver injury, AdipoR1, AMPKα

## Abstract

Radiation liver injury is a common complication of hepatocellular carcinoma radiotherapy. It is mainly caused by irreversible damage to the DNA of hepatocellular cells directly by radiation, which seriously interferes with metabolism and causes cell death. AdipoRon can maintain lipid metabolism and stabilize blood sugar by activating adiponectin receptor 1 (AdipoR1). However, the role of AdipoRon/AdipoR1 in the regulation of ionizing radiation (IR)-induced mitochondrial damage remains unclear. In this study, we aimed to elucidate the roles of AdipoRon/AdipoR1 in IR-induced mitochondrial damage in normal hepatocyte cells. We found that AdipoRon treatment rescued IR-induced liver damage in mice and mitochondrial damage in normal hepatocytes in vivo and in vitro. AdipoR1 deficiency exacerbated IR-induced oxidative stress, mitochondrial dynamics, and biogenesis disorder. Mechanistically, the absence of AdipoR1 inhibits the activity of adenosine monophosphate-activated protein kinase α (AMPKα), subsequently leading to disrupted mitochondrial dynamics by decreasing mitofusin (MFN) and increasing dynamin-related protein 1 (DRP1) protein expression. It also controls mitochondrial biogenesis by suppressing the peroxisome proliferator-activated receptor-gamma coactivator-1 alpha (PGC1α) and transcription factor A (TFAM) signaling pathway, ultimately resulting in impaired mitochondrial function. To sum up, AdipoRon/AdipoR1 maintain mitochondrial function by regulating mitochondrial dynamics and biogenesis through the AdipoR1-AMPKα signaling pathway. This study reveals the significant role of AdipoR1 in regulating IR-induced mitochondrial damage in hepatocytes and offers a novel approach to protecting against damage caused by IR.

## 1. Introduction

Mitochondria, which are the main sites where cells carry out aerobic respiration, have multiple functions. They are responsible for producing ATP and various biosynthetic intermediates. Additionally, mitochondria play a role in cellular stress responses, such as autophagy and apoptosis [1]. Mitochondria form a dynamic, interconnected network that is closely integrated with other cellular compartments. Mitochondrial dysfunction is increasingly recognized as a key component in both acute and chronic allostatic states [2]. Therefore, mitochondrial dysfunction has become a key factor in many diseases [3].

Radiotherapy has increasingly become an effective treatment for tumors; however, it will inevitably damage normal cells to some extent. The main target of IR damage is the double-stranded DNA in the nucleus. However, there are reports of the effects of radiation on cell organelles other than the nucleus [4]. Notably, mitochondria are the only sites where extranuclear DNA resides. IR can cause various lesions in the circular mitochondrial DNA, such as strand breaks, base mismatches, and large deletions, which are also observed in nuclear DNA [5,6]. Therefore, in addition to the nucleus, mitochondria are likely to be the main target of ionizing radiation (IR) [7]. IR can damage mitochondrial functions [8], increase mitochondrial oxidative stress [9], and induce apoptosis [10]. IR causes specific changes in mitochondrial gene expression that are related to cell survival [11], and mitochondria have been reported as the primary target for radiation-induced apoptosis [12].

Adiponectin contains a highly conserved complement factor C1q-like globular domain that can bind to and activate the membrane receptors AdipoR1 and activate adiponectin receptor 2. The primary physiological action of adiponectin is to ameliorate insulin resistance [13]. It has been reported that adiponectin can regulate peroxisome proliferator-activated receptor-gamma coactivator-1 alpha (PGC1α) and mitochondria through the adenosine monophosphate-activated protein kinase α (AMPKα) pathway [14]. At the same time, ablation of AdipoR1 causes myocardial mitochondrial dysfunction [15]. Our previous study demonstrated that AdipoR1 serves as a potential prognostic marker in patients undergoing stereotactic radiotherapy, indicating its role in modulating radiation sensitivity. However, the underlying mechanism of AdipoR1 in IR-induced mitochondrial damage is unclear.

This study will explore the effect of AdipoR1 on IR-induced mitochondrial damage and provide an experimental basis for further studying its related mechanism.

## 2. Results

### 2.1. AdipoRon Treatment Reversed IR-Induced Mice Liver Injury

In order to explore the effect of AdipoRon on IR-induced liver injury in mice in vivo, we gave mice a tail vein injection of AdipoRon (Figure 1A). Western blotting and RT-qPCR results showed that after four tail vein injections of AdipoRon, the protein and mRNA levels of AdipoR1 in the liver of mice increased significantly compared with the control group (Figure 1B,C). Next, we injected AdipoRon into the tail vein of mice four times at 1.2 mg/kg and then irradiated the liver of mice with a single dose of 30 Gy at a dose rate of 3 Gy/min. After 10 days of irradiation, the mice were sacrificed to obtain blood and liver samples for subsequent experiments. The serum biochemical results of mice showed that, compared with the control group, aspartate aminotransferase/alanine aminotransferase (AST/ALT) in mice was significantly increased after IR, which suggested that IR induced liver damage in mice. However, AdipoRon treatment could reverse the ratio of AST to ALT in mouse serum (Figure 1D), indicating that AdipoRon could rescue IR-induced liver damage in mice to a certain extent. The results of H&E staining showed vacuolar degeneration and nuclear condensation of mouse liver cells after IR, suggesting IR-induced liver injury in mice. However, AdipoRon treatment could partially rescue the radiation-induced liver damage (Figure 1F). We then detected the expression of mitochondria-related proteins in the mouse liver, as shown in Figure 1E. Consistent with the in vitro results, AdipoRon treatment was able to offset the increase in PGC1α protein levels caused by IR and increase the expression of the mitochondrial fusion protein mitofusin2 (MFN2). The expression of translocase of outer mitochondrial membrane 20 (TOM20) was decreased after IR treatment, but AdipoRon treatment can also rescue the expression of TOM20, suggesting that AdipoRon could rescue the damage to mitochondrial membrane integrity caused by IR.

Thus, these results indicated that AdipoRon treatment was able to rescue the liver damage in mice caused by IR.

### 2.2. AdipoRon Treatment Reversed IR-Induced Mitochondrial Damage

We utilized AdipoRon to assess whether activating the adiponectin/AdipoR1 signaling pathway can mitigate IR-induced mitochondrial damage. The results showed that AdipoRon (2.5 μM) treatment inhibited IR-induced death of HHL-5 and LO-2 in normal hepatocytes (Figure 2A,B). Therefore, 2.5 μM was used to treat the hepatocytes in the subsequent experiments. We found that AdipoRon (2.5 μM) treatment alleviated mitochondrial oxidative stress (Figure 2C) and rescued the decrease in mitochondrial membrane potential (Figure 2D). The changes in mitochondrial membrane potential are related to mitochondrial membrane integrity; therefore, we examined alterations in mitochondrial membrane integrity. Our results showed that AdipoRon treatment reversed IR-induced damage to mitochondrial membrane integrity. It suppressed the increase in BAX protein expression and the decrease in mitochondrial membrane protein TOM20 induced by IR (Figure 2E). Next, we tested the related indicators of mitochondrial biogenesis and mitochondrial dynamics after AdipoRon and IR treatment. As shown in Figure 2F, AdipoRon can reverse the increase in mitochondrial biogenesis-related proteins TFAM and PGC1α caused by IR, suggesting that AdipoRon can ameliorate mitochondrial biogenesis disorders caused by IR stimulation. The expression of the fissile-related protein DRP1 was increased in the IR group, and treatment with AdipoRon could inhibit IR-induced mitochondrial division (Figure 2F).

These results indicated that the AdipoRon has a certain protective effect on IR-induced mitochondrial damage.

### 2.3. AdipoR1 Knockdown Aggravated the Damage of Mitochondrial Function Induced by IR

We found that IR can induce increased expression of AdipoR1 (Appendix A), which may be a cellular self-rescue response following exposure to IR. In order to further investigate the mechanism of AdipoR1 involved in IR-induced mitochondrial damage, we conducted follow-up experiments using two types of human normal liver cells, HHL-5 and LO-2. We designed three si-RNA sequences to inhibit the expression of AdipoR1, and the third siRNA significantly reduced AdipoR1 expression (Figure 3A). To further elucidate the role of AdipoR1 in mitochondrial function, we conducted experiments using si-AdipoR1③ in the following studies. Reactive oxygen species (ROS) analysis revealed a significant increase in ROS levels in the si-AdipoR1 + IR group (Figure 3B,C). Moreover, mitochondrial membrane potential significantly decreased in the si-AdipoR1 + IR group (Figure 3D). The expression of TOM20, Bcl2, and BAX was used to evaluate mitochondrial integrity. The results indicated that treatment with IR combined with AdipoR1 knockdown significantly reduced the expression of TOM20 and the ratio of Bcl2 to BAX (Figure 3E). Finally, we found that the knockdown of AdipoR1 exacerbated IR-induced cellular energy deficit (Figure 3F,G).

These results suggested that AdipoR1 knockdown combined with IR treatment aggravated mitochondrial damage.

### 2.4. AdipoR1 Silencing Exacerbated Mitochondrial Biogenesis and Disturbances in Mitochondrial Dynamics

Mitochondrial dynamics and biogenesis play a crucial role in maintaining the health of mitochondrial networks. Next, we tested the effect of AdipoR1 knockdown on mitochondrial biogenesis after IR treatment. The results indicated that the expression of TFAM and PGC1α increased after IR treatment. Interestingly, in the si-AdipoR1 + IR group, the expression of TFAM and PGC1α decreased significantly (Figure 4A). The mtDNA copy numbers also fit this trend (Figure 4B). These results indicate that knocking down AdipoR1 increased disturbances in mitochondrial biogenesis induced by IR. Previous studies have suggested that the disturbance of the mitochondrial fusion-fission balance plays a crucial role. Therefore, the proteins associated with mitochondrial dynamics, DRP1 and the mitochondrial fusion protein MFN, were detected. As shown in Figure 4C,D, AdipoR1 knockdown further downregulated mitofusin1 (MFN1) and MFN2 expression, and markedly upregulated the mitochondrial fission-associated protein DRP1 expression compared with the NC + IR group. These results indicated that AdipoR1 knockdown exacerbated IR-induced mitochondrial dynamics disorder.

### 2.5. AdipoR1 Knockdown Induced Mitochondrial Damage by Regulating p-AMPKα

As shown in Figure 5A, the expression of p-AMPKα was significantly reduced in HHL-5 cells after AdipoR1 knockdown. Therefore, we speculated that AdipoR1 regulates mitochondrial function through the AMPKα pathway. We used Compound C (CC), an AMPKα activity inhibitor, to treat HHL-5 cells. As shown in Figure 5B, the expression of p-AMPKα was significantly reduced by CC efficiently at a concentration of 5 μM. Considering the tolerance of cells to CC, we conducted CCK-8 experiments after treating them with various concentrations of CC. The results showed that there was no significant difference in cell activity between the 5 μM group and the control group, suggesting that this concentration had no inhibitory effect on cell viability (Figure 5C). Thus, in the following research, a concentration of 5 μM was chosen for CC treatment. ROS analysis also revealed that CC treatment significantly increased ROS production induced by IR treatment compared to the DMSO + IR group (Figure 5D,E). Furthermore, mitochondrial membrane potential analysis showed that the MMP significantly decreased in the CC + IR group (Figure 5F). Detection of mitochondrial integrity indicators also showed that IR combined with CC treatment induced a significant decrease in the expression of TOM20 and the ratio of Bcl2 to BAX (Figure 5G). Then, we found that the ATP levels decreased markedly in the CC + IR group (Figure 5H,I). These results suggested that inhibiting p-AMPKα combined with IR aggravated mitochondrial damage in HHL-5 and LO-2 cells.

### 2.6. Inhibition of p-AMPKα Activity Exacerbated Mitochondrial Biogenesis and Mitochondrial Dynamic Imbalance After IR

Next, we tested the effect of CC on mitochondrial biogenesis after IR. The results indicated that after CC treatment combined with IR, the expression of TFAM and PGC1α decreased significantly (Figure 6A). The mtDNA copy numbers also followed this trend (Figure 6B). As shown in Figure 6C,D, CC combined with IR treatment further downregulated MFN expression, and markedly upregulated DRP1 expression compared with the DMSO + IR group. These results indicate that inhibiting p-AMPKα exacerbates IR-induced mitochondrial biogenesis and mitochondrial dynamics disorder.

### 2.7. Activating AMPKα Could Rescue the Mitochondrial Damage Caused by AdipoR1

To confirm whether AdipoR1’s regulation of mitochondrial function depends on the AMPKα pathway, we treated AdipoR1 knockdown cells with Metformin, an agonist of AMPKα activity. As shown in Figure 7A, after treating HHL-5 cells with 2 mM Metformin, the expression of p-AMPKα significantly increased. ROS analysis revealed that mitochondrial ROS overload induced by AdipoR1 knockdown combined with IR treatment could be mitigated by Metformin (Figure 7B,C). After Metformin treatment, the decrease in MMP was also reversed (Figure 7D). Then, we assessed mitochondrial integrity by examining the expression of TOM20, Bcl2, and BAX. The results indicated that Metformin treatment could reverse the expression levels of TOM20 and the ratio of Bcl2 to BAX (Figure 7E). The ATP assay demonstrated that the energy shortage induced by AdipoR1 knockdown combined with IR treatment was alleviated by Metformin (Figure 7F,G). These results suggested that activating AMPKα could reverse the mitochondrial damage induced by AdipoR1 knockdown combined with IR treatment.

The effect of Metformin on mitochondrial biogenesis was also observed after knocking down AdipoR1 in combination with IR. The results indicated that the decreases in TFAM and PGC1α expression were reversed by Metformin treatment (Figure 7H). The mtDNA copy number also followed this trend (Figure 7I). In Figure 7H, the expression of MFN1 and MFN2 was increased in the IR and Metformin combined treatment (NC + IR + Met) group compared to the NC + IR group, while the expression of DRP1 was inhibited. The expression levels of MFN2 and MFN1 in the si-AdipoR1 + Met + IR group were similar to those in the NC + IR group, indicating that the changes in the mitochondrial fusion proteins caused by AdipoR1 knockdown were reversed by Metformin. The expression levels of DRP1 in the si-AdipoR1 + Met + IR group were also reduced by Metformin. These results demonstrate that activating AMPKα alleviated AdipoR1 knockdown and IR-induced mitochondrial biogenesis and mitochondrial dynamics disorder.

## 3. Discussion

Radiation therapy has increasingly become an effective treatment for tumors, but it also inevitably damages normal cells. Mitochondria, as the only organelle outside the nucleus where DNA exists, have become one of the targets of IR. In recent years, studies have also reported that adiponectin/AdipoR1 also plays an important role in mitochondrial biogenesis [16]. However, the effect of AdipoR1 on IR-induced mitochondrial damage is unclear.

One of the main mechanisms of IR damage is to increase the level of ROS in the body through the radiation products of water [17], resulting in oxidative stress damage, and mitochondria are the main site of ROS generation. Our results showed that IR could directly act on mitochondria, resulting in a decrease in membrane potential and ATP levels. Moreover, inhibition of AdipoR1 expression combined with IR treatment could further aggravate mitochondrial damage from IR. Mitochondrial dysfunction is reflected in both mitochondrial biogenesis and mitochondrial dynamics [18]. Our results indicated that IR induces a significant increase in mitochondrial copy number, whereas mitochondrial mtDNA copy number is suppressed after AdipoR1 knockdown. Mitochondria contain multiple copies of their genome [19], and proper control of mitochondrial DNA copy number is thought to be important for normal cellular function. The increase in mtDNA copy number following radiation stimulation, termed “mitochondrial polyploidization” [20], is thought to be a compensatory mechanism or adaptive response of mitochondria to maintain the function of post-irradiated cells and malignantly transformed progeny, which makes these cells survive after radiation exposure [8,21]. However, it is still a matter of debate whether the increase in mtDNA copy number after irradiation is beneficial. The increase in mtDNA content after irradiation may lead to the overproduction of mitochondrial-encoded subunits. Likewise, our results indicated that the expression of mitochondrial biogenesis-related proteins PGC1α and TFAM was increased after IR treatment, whereas the expression of these mitochondrial biogenesis-related proteins was suppressed after AdipoR1 knockdown. PGC1α is a transcriptional coactivator critical for mitochondrial biosynthesis that activates genes that regulate energy homeostasis and metabolism. In this context, PGC1α alters the metabolic rate and expression of genes involved in gluconeogenesis, fat oxidation, and mitochondrial biosynthesis [22].

Mitochondria are important dynamic organelles within the cytoplasm, showing continuous movement, fusion, and fission to form the mitochondrial network [23]. Defects in mitochondrial dynamics and function may lead to serious diseases such as Alzheimer’s disease, Parkinson’s disease, Huntington’s disease, and optic atrophy [24,25,26,27]. Mitochondrial fusion and fission are controlled by the GTPase family. For mitochondrial fission, DRP1 is recruited from the cytoplasm to the mitochondrial outer membrane [28]. The mediators of mitochondrial fusion are MFN1, MFN2, and optic atrophy 1 proteins [29]. Targeted cellular IR induces mitochondrial fission and a corresponding dysfunction of the mitochondrial respiratory chain. Therefore, the balance between fission and fusion events may be an adaptive stress response necessary for normal cellular function. Our results suggest that IR induces increased mitochondrial fission while inhibiting mitochondrial fusion, which is further enhanced by the knockdown of AdipoR1 expression, suggesting that Adior1 has an important regulatory role in IR-induced defects in mitochondrial dynamics.

To further investigate the mechanism of AdipoR1 in regulating mitochondrial damage after IR, we examined changes in its downstream target genes following AdipoR1 knockdown. Previous studies reported that AdipoR1 could regulate mitochondrial biogenesis via the AMPKα-PGC1 pathway [30,31,32], thus we detected the expression levels of AMPKα and p-AMPKα after the knockdown of AdipoR1. The results showed that the knockdown of AdipoR1 did not affect the expression of AMPKα but inhibited the active form of AMPKα, which downregulated the expression of p-AMPKα. Therefore, we used the AMPKα activity inhibitor compound C to inhibit the activity of AMPKα for subsequent experiments. Our results showed that the corresponding detection indicators, such as mitochondrial function, mitochondrial biogenesis, and mitochondrial dynamics, after inhibition of AMPKα activity were consistent with the results after inhibition of AdipoR1, so we suspected that AdipoR1 regulates mitochondrial damage induced by IR and is mediated by AMPKα.

AMPK plays a major role in regulating cellular energy balance. AMPKα is activated by an increase in AMP/ADP relative to ATP in response to changes in intracellular adenine nucleotide levels. In addition to its role in maintaining intracellular energy homeostasis, AMPKα also regulates whole-body energy metabolism [32,33]. Previous studies have shown that the regulation of AMPKα by adiponectin receptor 1 is mainly based on two pathways. One is that AdipoR1 induces extracellular Ca^2+^ influx and subsequently activates Ca^2+^/calmodulin-dependent protein kinase [34]. Another is that AdipoR1 activates AMPKα activity through liver kinase B1, which activates AMPKα by promoting the phosphorylation of the Thr172 site on the AMPKα subunit and enhancing the phosphorylation level of AMPKα [35]. The regulatory effect of activated AMPKα on mitochondrial biogenesis is mainly regulated by the Sirt1-PGC1α pathway [36,37]. However, in this study, the pathway through which AdipoR1 activates AMPKα needs to be further explored.

Similarly, studies have shown that AMPKα also has a regulatory role in mitochondrial dynamics, and AMPKα can inhibit mitochondrial fission protein DRP1 to reduce mitochondrial fission [38]. This is also consistent with our findings. Interestingly, it has been shown that impaired mitochondrial bioenergetics induce AMPKα activation, but this AMPKα response is also required for mitochondrial fission [39]. Taken together, these studies demonstrated that AMPKα activation was a key regulator of mitochondrial fission, but whether AdipoR1 could directly regulate mitochondrial dynamics and the mechanism of how AdipoR1-AMPKα regulates mitochondrial dynamics still needs to be further explored.

This study revealed a new mechanism affecting the radiation injury of human embryonic hepatocytes from the aspect of the regulation of mitochondrial function by AdipoRon/AdipoR1 and provided a new target or new idea for the protection of radiation injury.

## 4. Materials and Methods

### 4.1. Reagents and Antibodies

AdipoRon (AG-CR1-0154-M050) was obtained from AdipoGen (San Diego, CA, USA). Compound C (S7840) and Metformin (S5958) were obtained from Selleck Chemical (Houston, TX, USA). A protease inhibitor cocktail No. 11836 153 001 was from Roche Diagnostics (Basel, Switzerland). Primary antibody AdipoR1 (#518030) was purchased from Santa Cruz Biotechnology (Dallas, TX, USA). Antibodies Bcl2 (#3498S), BAX (#2772), GAPDH (#5174S), TFAM (#7495S), PGC1α (#2178S), DRP1 (#5391S), and MFN2 (#9482S) were obtained from Cell Signaling Technology (Danvers, MA, USA). MFN1 (#ab129154) was purchased from Abcam (Cambridge, UK), and anti-actin was acquired from Sigma-Aldrich (St. Louis, MO, USA). No. A3853. Secondary antibodies, goat anti-rabbit IgG (H + L) HRP conjugate (cat. No. 170-6515) and goat anti-mouse IgG (H + L) HRP conjugate (cat. No. 170-6516) were obtained from Bio-Rad Laboratories (Mississauga, ON, Canada).

### 4.2. Cell Culture

HHL-5 cells were donated by Professor Yujuan Shan, School of Public Health, Wenzhou Medical University. LO-2 cells were obtained from the Chinese Academy of Sciences cell bank in Beijing, China. All cells were routinely cultured in Dulbecco’s Modified Eagle’s Medium (DMEM) (Sigma-Aldrich, USA) containing 10% fetal bovine serum (Solarbio, Beijing, China) at 37 °C in a humidified atmosphere with 5% CO_2_.

### 4.3. Irradiation

An X-ray generator (X-RAD 320, Precision X-ray Inc., Madison, CT, USA) was used to administer radiation. Irradiation conditions: 20 kV, 12.5 mA, filter 1, distance (SSD) 50 cm, dose rate 300 cGy/min. The single dose was 6 Gy in cells or 30 Gy in mice [40].

### 4.4. Adenosine Triphosphate (ATP) Assay

The cellular ATP levels were detected using an ATP Bioluminescence Assay Kit (Promega, Madison, WI, USA). Cells were seeded in 96-well plates and left overnight. After treatment, cells were lysed, and the supernatants were collected. A working solution (50 μL) was added to 50 μL of the sample in the 96-well plate, and the luminescence was measured immediately using an automated microplate reader.

### 4.5. Analysis of Mitochondrial Reactive Oxygen Species (mtROS)

Mitochondrial ROS (mtROS) levels in HHL-5 cells were assessed using the MitoSOX Red (Invitrogen M36008, Waltham, MA, USA) assay following the manufacturer’s instructions. The cells were seeded in 6-well plates at a density of 2.5 × 10^4^/mL with 3 parallel wells in each group. After various treatments, cells were stained with a MitoSOX Red probe at a final concentration of 10 μM at 37 °C in the dark for 15 min. Afterward, the cells were washed with PBS three times, and the mtROS activity was analyzed using a flow cytometer (ACEA NovoCyte 2040R, Santa Clara, CA, USA).

### 4.6. Analysis of Mitochondrial Membrane Potential

The cells were seeded in 6-well plates with a density of 2 × 10^4^/mL with 3 parallel wells in each group. The cells were washed with PBS three times and then incubated to avoid light with JC-1 working solution at 37 °C for 20 min. After being washed with PBS three times, the images were captured by a fluorescence microscope (Nikon, Tokyo, Japan). Red staining indicates polarized mitochondria in JC-1 staining. Green staining indicates depolarized mitochondria in JC-1 staining.

### 4.7. Western Blotting

Cells were lysed in RIPA lysis buffer (Solarbio, Beijing, China) supplemented with protease and phosphatase inhibitors (Applygen, Beijing, China). Protein samples were subjected to 12% SDS-PAGE based on the protein mass and then transferred to PVDF membranes (Bio-Rad, Hercules, CA, USA). After the transfer was completed, the membranes were blocked with 5% nonfat milk in TBS/0.1% Tween 20 for 120 min. Incubation with the primary antibody (as indicated) was conducted overnight at 4 °C. Incubation with a peroxidase-conjugated anti-mouse or anti-rabbit secondary antibody (1:10,000) was performed for 120 min at room temperature. Finally, a chemiluminescent analysis was performed.

### 4.8. Transfection

Transfection with siRNA was used to inhibit AdipoR1 expression in HHL-5 and LO-2 cells. The siRNA targeting AdipoR1 and the negative control siRNA were purchased from Santa Cruz Biotechnology (Dallas, TX, USA). To transfect siRNA into HHL-5 cells and LO-2 cells, Opti-Minimal Essential Medium (Sigma-Aldrich, USA) without serum or antibiotics was used to incubate the cells for 24 h. Lipofectamine^®^ 2000 transfection reagent (Invitrogen; Thermo Fisher Scientific, Inc., Waltham, MA, USA) was added to the medium following the manufacturer’s protocol. siRNAs (70 nmol/L/well of siRNA) were subsequently added to the serum-free medium and incubated with cells for 72 h. The knockdown efficiency was evaluated using Western blot analysis.

### 4.9. Mitochondrial DNA (mtDNA) Copy Number

The mtDNA copy number was counted as described. After a 24 h ionizing radiation (IR) treatment, total DNA was extracted using the TIANamp Genomic DNA Kit (TIANGEN BIOTECH, Beijing, China). The mtDNA copy number was determined by qRT-PCR using TB Green^®^ Premix Ex Taq™ II (Tli RNaseH Plus) (Takara, Osaka, Japan) in a QuantStudio™ 3 Flex system (Applied Biosystems, Waltham, MA, USA). Thermal cycling conditions included an initial hold period at 95 °C for 5 min, followed by a two-step PCR program of 95 °C for 15 s and 60 °C for 40 s for a total of 35 cycles. The primers used in this study are listed below.

The ratio of mtDNA to nDNA was calculated as the relative mitochondrial DNA copy number. Sequence for primers: mtDNA-F: 5′-CACC CAAGAACAGGGTTTGT-3′; mtDNA-R: 5′-TGGCCATGGGTATGTTGTTAA-3′;18S-F: 5′-TAGAGGGACAAGTGGCGTTC-3′; 18S-R: 5′-CGCTGAGCCAGTCAGT GT-3′.

### 4.10. Cell Viability by Cell Counting Kit-8 (CCK-8) Assays

CCK-8 was used according to the manufacturer’s protocol. Cells were seeded in 96-well plates (3 × 10^3^ cells/well) and pretreated with CC (5, 10 and 15 μM). The OD values were recorded after 1 h of CCK-8 incubation at 450 nm using a plate reader. The proliferation rate of cells was calculated using the following formula: cell viability = OD (experimental group)/OD (control group) × 100%.

### 4.11. Mice and Treatment

Eight-week-old male BALB/c mice were obtained from Beijing Vital River Laboratory Animal Technology. The animals were housed under controlled, pathogen-free conditions. AdipoRon (10 mg) (Adipogen, San Diego, CA, USA) was reconstituted with 500 µL of DMSO (stock) and stored at −20 °C. The animals were given either a single intravenous dose of AdipoRon 1.2 mg/kg BW in PBS (*n* = 5) or an equivalent volume of DMSO in PBS (*n* = 3, Controls) in a total volume of 100 µL via tail vein injections once every other day 4 times.

### 4.12. Statistical Analysis

All statistical analyses were performed using GraphPad Prism 8 software. The results were expressed as means ± standard deviation. Student’s *t*-test was used to compare the difference between the two groups. For comparisons between multiple groups, an ordinary one-way or two-way ANOVA was used. *p* values less than 0.05 *, 0.01 **, or 0.001 *** were considered statistically significant.

## 5. Conclusions

To sum up, AdipoRon protects the liver from ionizing radiation-induced damage by enhancing AdipoR1, and AdipoR1 maintains mitochondrial function by regulating mitochondrial dynamics and biogenesis through the AdipoR1-AMPKα signaling pathway. This study uncovers the important role of AdipoR1 in regulating IR-induced mitochondrial damage to hepatocytes and provides a new approach to protecting against the damage caused by IR.

## Figures and Tables

**Figure 1 ijms-25-11277-f001:**
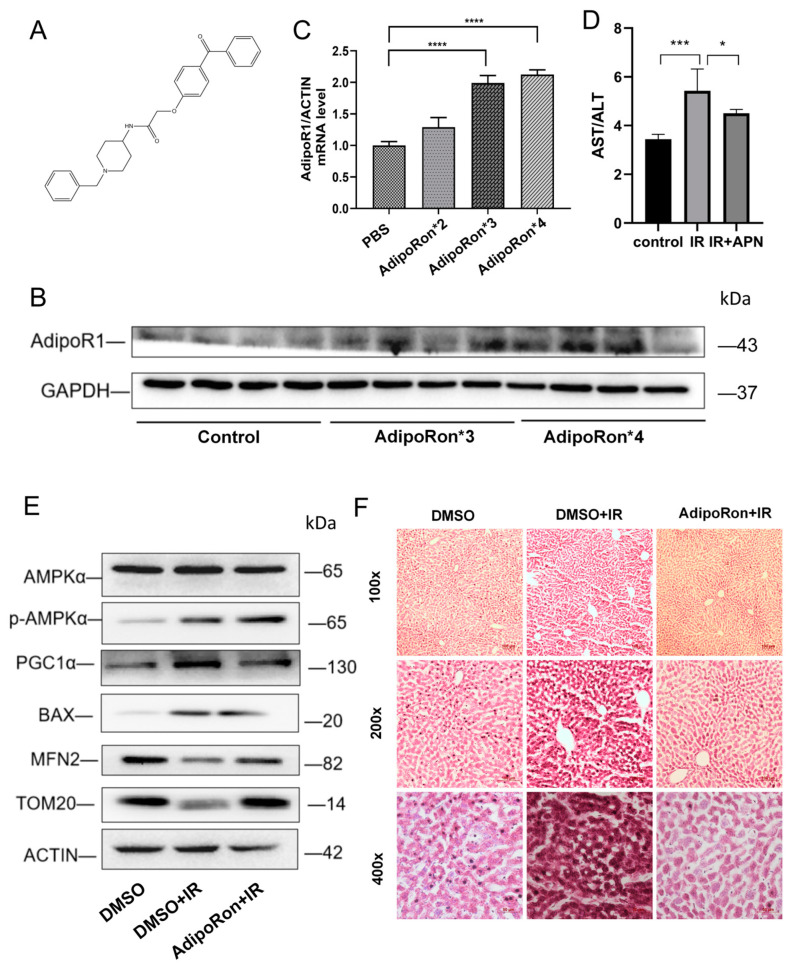
AdipoRon reversed IR-induced mouse liver injury. (**A**) Chemical structure of AdipoRon. (**B**) Protein expression of AdipoR1 in mice liver after different doses of tail vein injection of AdipoRon. (**C**) mRNA expression of AdipoR1 in mice liver after different doses of tail vein injection of AdipoRon. (**D**) The ratio of ALT and AST in the blood of mice in different treatment groups. (**E**) Western blot analysis of AMPKα, p-AMPKα, TOM20, MFN2, PGC1α, and BAX expression levels in different groups. (**F**) H&E staining of mice livers in different groups. Data represent the mean ± SEM (*n* = 3 independent repeats); *p* values were calculated using one-way ANOVA. * *p* < 0.05, *** *p* < 0.001 and **** *p* < 0.0001.

**Figure 2 ijms-25-11277-f002:**
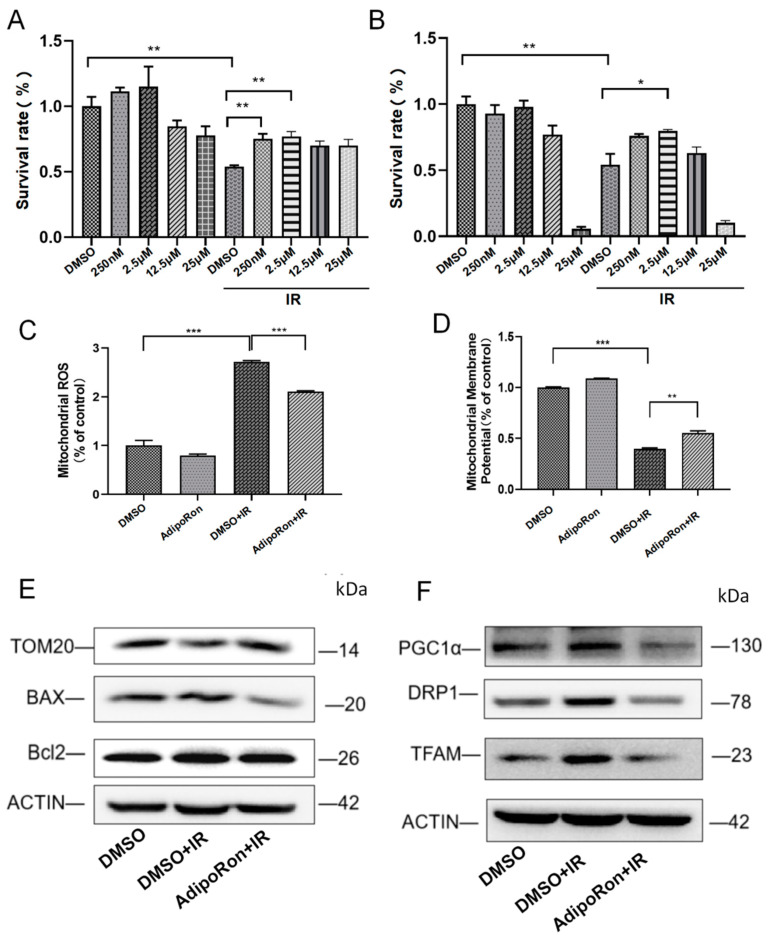
AdipoRon reversed IR-induced mitochondrial damage. (**A**) HHL-5 cells were pretreated with different concentrations of AdipoRon. Cell viability was detected by the CCK-8 kit. (**B**) LO-2 cells were pretreated with different concentrations of AdipoRon. The cell viability was detected by the CCK-8 kit. (**C**) Mitochondrial reactive oxygen species levels were analyzed by the mitoSOX kit in HHL-5 cells. (**D**) Mitochondrial membrane potential levels were analyzed by DIOC6 staining in HHL-5 cells. (**E**) Western blot analysis of TOM20, Bcl2, and BAX expression levels in HHL-5 cells. (**F**) Western blotting was used to analyze the changes in PGC1α, TFAM, and DRP1. Data represent the mean ± SEM (*n* = 3 independent repeats); *p* values were calculated using one-way ANOVA. * *p* < 0.05, ** *p* < 0.01, and *** *p* < 0.001.

**Figure 3 ijms-25-11277-f003:**
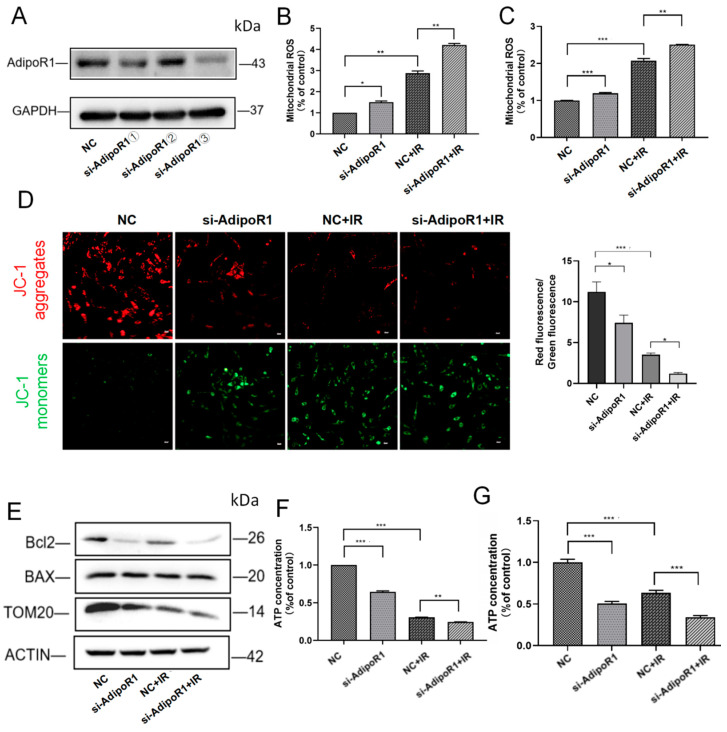
AdipoR1 knockdown exacerbated mitochondrial function damage after IR. (**A**) Western blot analysis of AdipoR1 levels in HHL-5 cells following AdipoR1 knockdown treatment. (**B**) Mitochondrial ROS levels in HHL-5 cells were analyzed using the mitoSOX kit. (**C**) Mitochondrial ROS levels in LO-2 cells were analyzed using the mitoSOX kit. (**D**) After treatment, the cells were subjected to JC-1 staining for mitochondrial membrane potential assessment. Red staining indicates polarized mitochondria in JC-1 staining. Green staining indicates depolarized mitochondria in JC-1 staining. Scale bar: 500 μm. (**E**) Expression of the proteins TOM20, Bcl2, and BAX in HHL-5 cells was demonstrated by Western blot analysis. (**F**) ATP content in different groups of HHL-5 cells. (**G**) ATP content in different groups of LO-2 cells. Data represent the mean ± SEM (*n* = 3 independent repeats). *p* Values were calculated using one-way ANOVA. * *p* < 0.05, ** *p* < 0.01, and *** *p* < 0.001.

**Figure 4 ijms-25-11277-f004:**
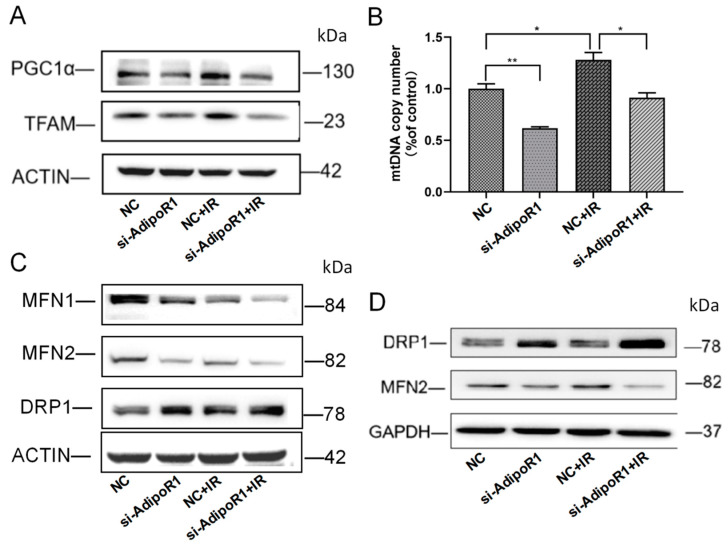
AdipoR1 knockdown exacerbated mitochondrial biogenesis and disrupted mitochondrial dynamics after IR. (**A**) Western blotting was used to analyze the changes in PGC1α and TFAM in HHL-5 cells. (**B**) Relative mtDNA copy number in HHL-5 cells. (**C**) The expressions of MFN1, MFN2, and DRP1 in HHL-5 cells were detected using Western blot analysis. (**D**) The expressions of MFN2 and DRP1 in LO-2 cells were detected using Western blot analysis. Data represent the mean ± SEM (*n* = 3 independent repeats). *p* Values were calculated using one-way ANOVA. * *p* < 0.05, ** *p* < 0.01.

**Figure 5 ijms-25-11277-f005:**
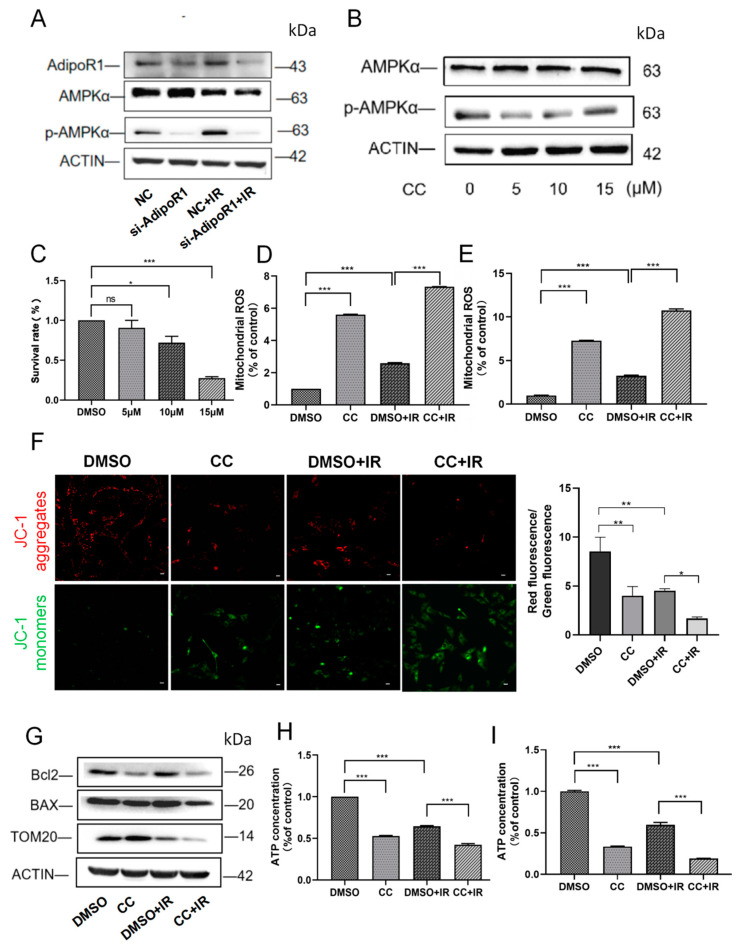
Inhibiting p-AMPKα led to more severe mitochondrial damage following IR treatment. (**A**) Western blot analysis of AMPKα and p-AMPKα levels in HHL-5 cells following AdipoR1 knockdown. (**B**) Western blot analysis was conducted to assess the levels of AMPKα and p-AMPKα in HHL-5 cells after treatment with CC. (**C**) HHL-5 cells were pretreated with various concentrations of Compound C. Cell viability was assessed using the CCK-8 kit. (**D**) Mitochondrial reactive oxygen species levels in HHL-5 cells were analyzed using the mitoSOX kit. (**E**) Mitochondrial reactive oxygen species levels in LO-2 cells were analyzed using the mitoSOX kit. (**F**) After treatment, the HHL-5 cells were subjected to JC-1 staining to assess mitochondrial membrane potential. Red staining indicates polarized mitochondria in JC-1 staining. Green staining indicates depolarized mitochondria in JC-1 staining. Scale bar: 500 μm. (**G**) The expression of the proteins TOM20, Bcl2, and BAX in HHL-5 cells was demonstrated by Western blot analysis. (**H**) ATP content in different groups of HHL-5 cells. (**I**) ATP content in different groups of LO-2 cells. Data represent the mean ± SEM (*n* = 3 independent repeats). *p* Values were calculated using one-way ANOVA. * *p* < 0.05, ** *p* < 0.01, and *** *p* < 0.001.

**Figure 6 ijms-25-11277-f006:**
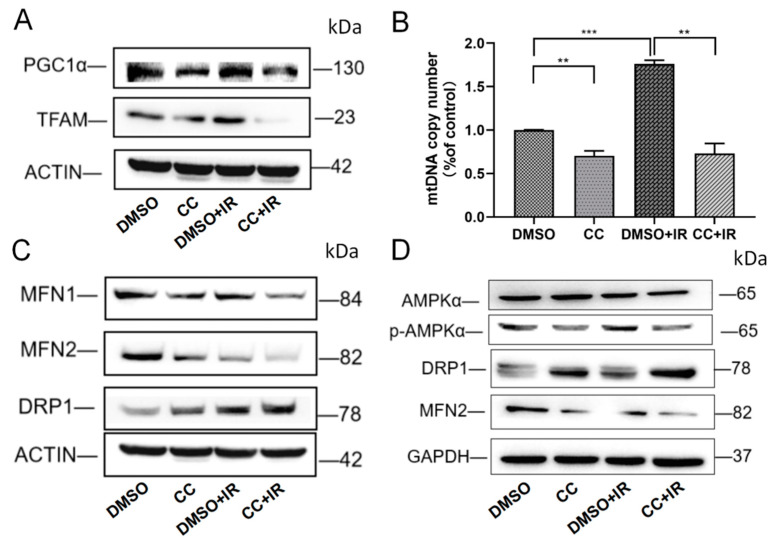
Inhibition of p-AMPKα activity exacerbated mitochondrial biogenesis and mitochondrial dynamic imbalance after IR. (**A**) Western blotting was used to analyze the changes in PGC1α and TFAM in HHL-5 cells. (**B**) Relative mtDNA copy number in HHL-5 cells. (**C**) The expression of the proteins MFN1, MFN2, and DRP1 was detected by Western blotting in HHL-5 cells. (**D**) The expression of the proteins AMPKα, p-AMPKα, MFN2, and DRP1 was detected using Western blot analysis in LO-2 cells. Data represent the mean ± SEM (*n* = 3 independent repeats). *p* Values were calculated using one-way ANOVA. ** *p* < 0.01, and *** *p* < 0.001.

**Figure 7 ijms-25-11277-f007:**
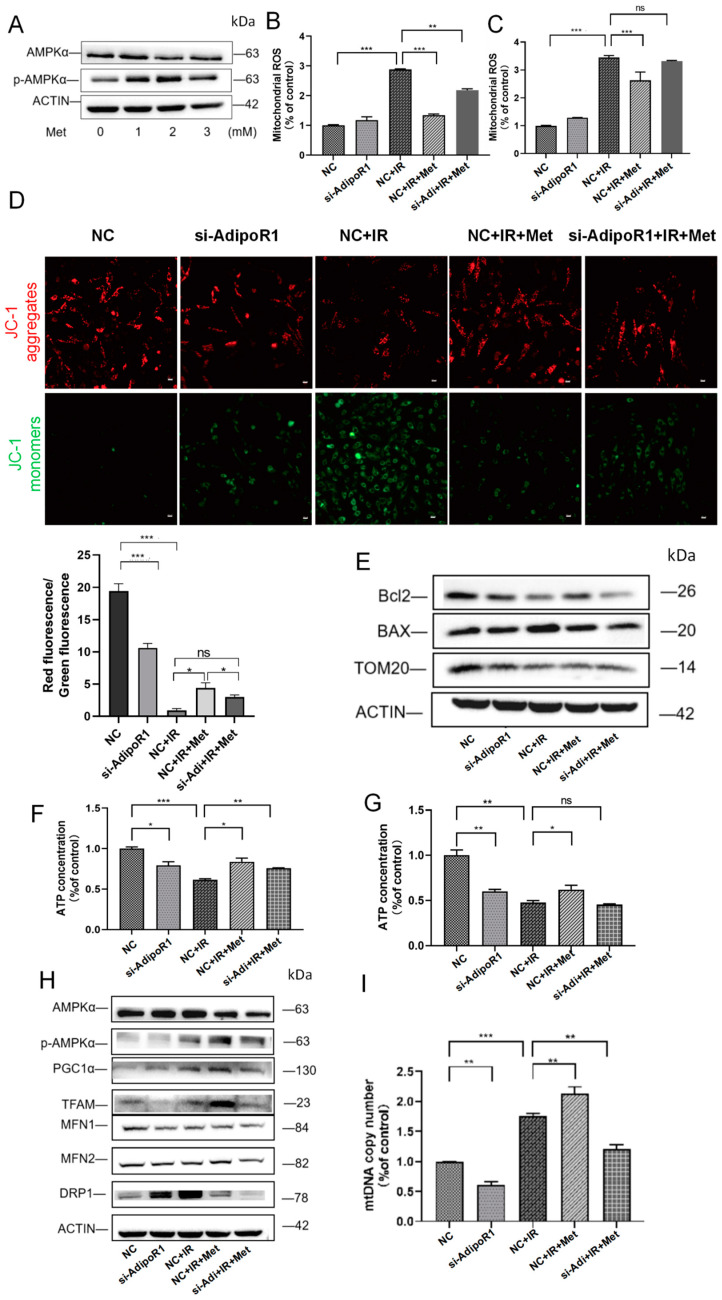
Activating p-AMPKα alleviated the damage to mitochondrial function following AdipoR1 knockdown combined with IR. (**A**) Western blot analysis of AMPKα and p-AMPKα levels in HHL-5 cells following Metformin treatment. (**B**) Mitochondrial reactive oxygen species levels were analyzed in HHL-5 cells using the mitoSOX kit. (**C**) Mitochondrial reactive oxygen species levels were analyzed in LO-2 cells using the mitoSOX kit. (**D**) After treatment, the cells were subjected to JC-1 staining to assess mitochondrial membrane potential. Red staining indicates polarized mitochondria in JC-1 staining. Green staining indicates depolarized mitochondria in JC-1 staining. Scale bar: 500 μm. (**E**) The expression of the proteins TOM20, Bcl2, and BAX in HHL-5 cells was demonstrated by Western blot analysis. (**F**) ATP content in different groups of HHL-5 cells. (**G**) ATP content in different groups of LO-2 cells. (**H**) The expression of the proteins AMPKα, p-AMPKα, PGC1α, TFAM, MFN1, MFN2, and DRP1 in HHL-5 cells was detected using Western blot analysis. (**I**) Relative mtDNA copy number in HHL-5 cells. Data represent the mean ± SEM (*n* = 3 independent repeats). *p* Values were calculated using one-way ANOVA. * *p* < 0.05, ** *p* < 0.01, and *** *p* < 0.001.

## Data Availability

All data generated or analyzed during this study are included in this article.

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
