# Peer review of "AdipoRon Alleviates Liver Injury by Protecting Hepatocytes from Mitochondrial Damage Caused by Ionizing Radiation"

_ijms, 2024, doi:10.3390/ijms252011277_

Round 1

Reviewer 1 Report

Comments and Suggestions for Authors

The manuscript entitled, “AdipoRon alleviates liver injury by protecting hepatocytes from mitochondrial damage caused by ionizing radiation” demonstrates that AdipoRon protects liver from radiation damage by reducing mitochondrial ROS and enhancing mitochondrial biogenesis. Specifically, AdipoRon enhances ADIPOR1 which activates AMPK that enhances mitochondrial biogenesis and reduces mitochondrial ROS. Overall, this study has been conducted appropriately and the data are convincing, and the authors have identified a specific pathway that AdipRon can protect from radiation damage of hepatocytes. However, there are some things that need to be done to improve this manuscript.  Below are items that need to be addressed to improve this manuscript:

1.       Results section 2.1, the mouse model of AdipoRon administration needs to be stated clearly (this was eventually found in the materials and methods section, but don’t make your readers hunt for this information): what dose of AdipoRon was used, when specifically, was it administered? How much radiation was administered? How was this targeted to the liver? Why was 30 Gy chosen, this is not therapeutically relevant dose. Please give a rationale.

2.       Was AdipoR2 investigated? Was it altered with AdipoRon treatment? Why was only AdipoR1 investigated as AdipoRon binds to both receptors?

3.       For all Figures with westernblots, they need to be quantified using densitometry. Statements cannot be made that protein levels were significantly enhanced with one picture of a westernblot. Statistics need to be performed on densitometry from 3 independently run blots to make this statement.

4.       Figure 1F, it is not clear what is damaged in the livers of DMSO+IR treated mice. The pathology needs to be described in the results section. Also, how long after radiation exposure were the tissues harvested? This needs to be stated in the results or in the figure legend.

5.       Figure 2C, D, E, and F, what dose of AdipoRon was chosen for these studies?

6.       Figure 3D, JC-1 staining is a way to measure mitochondrial potential. Mitochondria staining red for aggregates means high membrane potential and healthy mitochondria. Conversely, mitochondria staining green have low membrane potential and are thought to be unhealthy. Why would you merge these images? To analyze this data correctly, quantify red and green staining and show ratio of these values.

7.       Figure 3D these images need to quantified to say that there is a significant change between groups. Magnification bars need to be added on these images to demonstrate that the same magnification was used when taking pictures of all groups. These steps should be done for all immunofluorescence images in the manuscript.

8.       Confocal imaging of mitochondria would add to this manuscript to determine if mitochondria are punctated or fused with the different conditions.

9.       Figure 7, what dose of metformin was used for the rest of the assays displayed in Figure 7?

Author Response

Comments 1: [Results section 2.1, the mouse model of AdipoRon administration needs to be stated clearly (this was eventually found in the materials and methods section, but don’t make your readers hunt for this information): what dose of AdipoRon was used, when specifically, was it administered? How much radiation was administered? How was this targeted to the liver? Why was 30 Gy chosen, this is not therapeutically relevant dose. Please give a rationale.]

Response 1: [(1)We injected AdipoRon into the tail vein of mice for 4 times at 1.2mg/kg and then irradiated the liver of mice with a single dose of 30Gy at a dose rate of 3Gy/min. It has added in lines 72-74 of Result 2.1. (2)According to the mice physiological structure, we customized a special lead mold to shield the mice with radiation except for the liver. (3)It is generally considered that the safe radiation tolerance dose of normal liver is 30-35 Gy, 45-47 Gy for two thirds of the liver, and 70-90 Gy for one third of the liver under conventional dose fractionation.(Partial volume tolerance of the liver to radiation.PMID: 16183482). It has been cited in line 349 of Materials and methods 4.3. For this dose of 30Gy, we are using a single radiation treatment. This is not a traditional dose of tumor therapy, but a treatment to establish a model of liver injury. ] 

Comments 2: [Was AdipoR2 investigated? Was it altered with AdipoRon treatment? Why was only AdipoR1 investigated as AdipoRon binds to both receptors?]

Response 2: 

[As we found that AdipoRon-treated irradiated cells showed altered mitochondrial function and biogenesis, AMPKα has been reported to be involved in this process. AdipoR1 can activate AMPKα, while AdipoR2 mainly activates PPARα

to participate in lipid metabolism, which was reported in Hepatic AdipoR2 signaling plays a protective role against progression of nonalcoholic steatohepatitis in mice (Hepatology. PMID: 18666257).The effects of AdipoR2 on regulating lipid metabolism were investigated in our another study (data was not shown). We therefore focused only on AdipoR1 in this manuscript .]

Comments 3: [For all Figures with westernblots, they need to be quantified using densitometry. Statements cannot be made that protein levels were significantly enhanced with one picture of a westernblot. Statistics need to be performed on densitometry from 3 independently run blots to make this statement.]

Response 3: 

[It is added in the Supplementary material.]

Comments 4: [Figure 1F, it is not clear what is damaged in the livers of DMSO+IR treated mice. The pathology needs to be described in the results section. Also, how long after radiation exposure were the tissues harvested? This needs to be stated in the results or in the figure legend.]

Response 4: 

[(1)The description of the pathological findings has been supplemented.The results of H&E staining showed vacuolar degeneration and nuclear condensation of mouse liver cells after ionizing radiation, suggesting that ionizing radiation-induced liver injury in mice. However, AdipoRon treatment could partially rescue the radiation-induced liver damage. It has added in lines 79-82 of Result 2.1. (2)After 10 days of irradiation, the mice were sacrificed to obtain blood and liver samples for subsequent experiments. It has added in lines 74-75 of Result 2.1 ]

Comments 5: [Figure 2C, D, E, and F, what dose of AdipoRon was chosen for these studies?]

Response 5: 

[Our carelessly caused a writing error, which has now been corrected. We used 2.5μM AdipoRon to treat cells. It has added in lines 105-107 of Result 2.2.]

Comments 6: [6.Figure 3D, JC-1 staining is a way to measure mitochondrial potential. Mitochondria staining red for aggregates means high membrane potential and healthy mitochondria. Conversely, mitochondria staining green have low membrane potential and are thought to be unhealthy. Why would you merge these images? To analyze this data correctly, quantify red and green staining and show ratio of these values.]

Response 6: 

[Thank you for your suggestion, it has been corrected in Fig.3D.]

Comments 7: [Figure 3D these images need to quantified to say that there is a significant change between groups. Magnification bars need to be added on these images to demonstrate that the same magnification was used when taking pictures of all groups. These steps should be done for all immunofluorescence images in the manuscript.]

Response 7: 

[It has been corrected in all immunofluorescence images in the manuscript.]

Comments 8: [Confocal imaging of mitochondria would add to this manuscript to determine if mitochondria are punctated or fused with the different conditions.]

Response 8: 

[ I am very sorry that due to the limitation of experimental conditions in our institution, we do not have a laser confocal microscope that can photograph the temporal dynamic changes of mitochondria, if there is a chance in the future, we will look for equipment to improve the experiment.]

Comments 9: [Figure 7, what dose of metformin was used for the rest of the assays displayed in Figure 7?]

Response 9: 

[Our carelessly caused a writing error, which has now been corrected. We used 2 mM metformin to treat cells. It has corrected in lines 245 of Result 2.7]

Reviewer 2 Report

Comments and Suggestions for Authors

The authors evaluated the role of AdipoR1 in regulating IR-induced mitochondrial damage in hepatocytes and offered a novel approach to protecting against damage caused by IR.

1) Please define all abbreviations at their first appearance to help the reader's understanding.

2) Could you quantify protein expression data from Western blotting? In additioin, the unit is "Da".

3) The results showed that AdipoRon (250nM) treatment inhibited IR-induced death of HHL-5 and LO-2 in normal hepatocytes (Fig. 2A,B). Isn't the correct value 2.5μM?

4) alleviated mitochondrial oxidative stress (Figure 2C), and rescued the decrease in mitochondrial membrane potential (Figure 2D). Which cell line is this data from?

5) It would be easier to understand if you could provide some representative image data on mitochondrial morphological changes.

6) As shown in Figure 5B, the expression of p-AMPKα was significantly reduced by Compound C efficiently at a concentration of 5 μM. Considering the tolerance of cells to CC, we conducted CCK-8 experiments after treating them with various concentrations of CC. The results showed that there was no significant difference in cell activity between the 5 μM group and the control group, suggesting that this concentration had no inhibitory effect on cell viability (Figure 5C). 5 μM is the compound that most strongly inhibits p-AMPKα expression and produces the most mitochondrial ROS, so why does it not affect viability?

7) As shown in Figure 7A, after treating HHL-5 cells with 200 μM metfor- 233 min, the expression of p-AMPKα significantly increased. Data are shown for 1, 2, and 3 mM only.

8) ROS analysis revealed that mitochondrial ROS overload induced by AdipoR1 knockdown combined with IR treatment could be mitigated by Metformin (Figure 7B and C).  Why are there no data for si-AdipoR1+IR?

Author Response

Comments 1: [Please define all abbreviations at their first appearance to help the reader's understanding.]

Response 1: [Thank you for your suggestion, the abbreviations have been added to the manuscript lines 477-483. ] 

Comments 2: [Could you quantify protein expression data from Western blotting? In additioin, the unit is "Da".]

Response 2: 

[ It is added in the Supplementary material. And units have been corrected in all Figures.]

Comments 3: [The results showed that AdipoRon (250nM) treatment inhibited IR-induced death of HHL-5 and LO-2 in normal hepatocytes (Fig. 2A,B). Isn't the correct value 2.5μM?]

Response 3: 

[Our carelessly caused a writing error, which has now been corrected. We used 2.5μM AdipoRon to treat cells. It has added in lines 105-107 of Result 2.2]

Comments 4: [alleviated mitochondrial oxidative stress (Figure 2C), and rescued the decrease in mitochondrial membrane potential (Figure 2D). Which cell line is this data from?]

Response 4: 

[We used HHL-5 cells for the experiments in Fig. 2C and D. It has added in line 128-130 of legend (Figure 2). ]

Comments 5: [ It would be easier to understand if you could provide some representative image data on mitochondrial morphological changes.]

Response 5: 

[ I am very sorry that due to the limitation of experimental conditions in our institution, we do not have a laser confocal microscope or TEM that can photograph the temporal dynamic and morphological changes of mitochondria, if we have a chance in the future, we will look for equipment to improve the experiment.]

Comments 6: [As shown in Figure 5B, the expression of p-AMPKα was significantly reduced by Compound C efficiently at a concentration of 5 μM. Considering the tolerance of cells to CC, we conducted CCK-8 experiments after treating them with various concentrations of CC. The results showed that there was no significant difference in cell activity between the 5 μM group and the control group, suggesting that this concentration had no inhibitory effect on cell viability (Figure 5C). 5 μM is the compound that most strongly inhibits p-AMPKα expression and produces the most mitochondrial ROS, so why does it not affect viability?]

Response 6: 

[Our CCK-8 experiment was conducted at 24h after drug treatment. This time did not reach the doubling cycle time of HHL-5 cells. So cells have not yet proliferated much. Therefore there was no significant inhibitory effect in the short term.]

Comments 7: [As shown in Figure 7A, after treating HHL-5 cells with 200 μM metfor- 233 min, the expression of p-AMPKα significantly increased. Data are shown for 1, 2, and 3 mM only.]

Response 7: 

[Our carelessly caused a writing error, which has now been corrected. We used 2 mM metformin to treat cells. It has corrected in lines 245 of Result 2.7]

Comments 8: [ROS analysis revealed that mitochondrial ROS overload induced by AdipoR1 knockdown combined with IR treatment could be mitigated by Metformin (Figure 7B and C). Why are there no data for si-AdipoR1+IR?]

Response 8: 

[ Figure 7 showed that compared with NC+IR group, ROS accumulation in NC+IR+Met group was significantly reduced, indicating that metformin had a rescue effect on ROS accumulation. Moreover, the accumulation of ROS was reduced or no significant difference was found in si-AdipoR1+IR+Met group compared with the NC+IR group.

As shown in Result 2.3, we have confirmed that ROS was significantly accumulated after si-AdipoR1+IR compared with the NC+IR group. Therefore we did not add the comparison of this group in Figure 7.]
